# Bioactive Compounds and Antioxidant Activity from Spent Coffee Grounds as a Powerful Approach for Its Valorization

**DOI:** 10.3390/molecules27217504

**Published:** 2022-11-03

**Authors:** Carolina Andrade, Rosa Perestrelo, José S. Câmara

**Affiliations:** 1CQM—Centro de Química da Madeira, Campus da Penteada, Universidade da Madeira, 9020-105 Funchal, Portugal; 2Departamento de Química, Faculdade de Ciências Exatas e Engenharia, Campus da Penteada, Universidade da Madeira, 9020-105 Funchal, Portugal

**Keywords:** antioxidant potential, bioactive compounds, spent coffee grounds, µ-SPEed/UHPLC-PDA

## Abstract

Coffee is one of the world’s most popular beverages, and its consumption generates copious amounts of waste. The most relevant by-product of the coffee industry is the spent coffee grounds, with 6 million tons being produced worldwide per year. Although generally treated as waste, spent coffee grounds are a rich source of several bioactive compounds with applications in diverse industrial fields. The present work aimed at the analysis of spent coffee grounds from different geographical origins (Guatemala, Colombia, Brazil, Timor, and Ethiopia) for the identification of bioactive compounds with industrial interest. For this purpose, the identification and quantification of the bioactive compounds responsible for the antioxidant activity attributed to the spent coffee grounds were attempted using miniaturized solid-phase extraction (µ-SPEed), combined with ultrahigh-performance liquid chromatography with photodiode array detection (UHPLC-PDA). After validation of the µ-SPEed/UHPLC-PDA method, this allowed us to conclude that caffeine and 5-caffeoylquinic acid (5-CQA) are the most abundant bioactive compounds in all samples studied. The total phenolic content (TPC) and antioxidant activity are highest in Brazilian samples. The results obtained show that spent coffee grounds are a rich source of bioactive compounds, supporting its bioprospection based on the circular economy concept closing the loop of the coffee value chain, toward the valorization of coffee by-products.

## 1. Introduction

Coffee is one of the most popular beverages worldwide. According to the International Coffee Organization, over 169 million 60 kg bags of coffee were produced in 2020, which represents a 0.3% increase from the previous year. With coffee production and consumption being expected to steadily grow in the next years, the amount of byproducts produced by the coffee industry is also projected to increase [1,2]. Several byproducts are generated during the processing that coffee undergoes until it reaches the final roasted beans, including the husk, pulp, mucilage, parchment, silver skin, and spent coffee grounds. Spent coffee grounds are the coffee industry’s final byproduct and the biggest contributor to coffee’s biowaste [3]. Over 6 million tons of spent coffee grounds are produced worldwide per year [4]. While spent coffee grounds are usually treated as waste, they can be a source of raw material for application in several industrial fields.

The implementation of adjusted management of spent coffee grounds is still needed. Most of this coffee byproduct ends up being disposed of in landfills, which represents a great environmental hazard. Without prior treatment, this practice leads to the emission of carbon dioxide and other greenhouse gases, as well as the release of chemical compounds (e.g., caffeine, polyphenols, and tannins) into the environment [5]. Another practice used in Europe for the management of this type of waste is incineration, which is also an environmental hazard [6]. Moreover, with climate change’s impacts being more and more aggravated, international organizations encourage the valorization of materials previously treated as waste [7]. Spent coffee grounds have been used for bioethanol and biodiesel generation, and in wastewater treatment (for instance, in the reduction of cationic dyes due to their adsorption properties) [8]. However, these applications are quite limited. Moreover, when it comes to biowaste valorization, the application of a biorefinery approach is favored, prioritizing the extraction of valuable compounds with potential applications in the industry [9]. Some of the bioactive compounds present in coffee are extracted when the beverage is prepared, but an important source of these compounds (including: chlorogenic, caffeic, caffeoylquinic, ellagic, trans-ferulic, feruloylquinic, gallic, p-hydroxybenzoic, p-coumaric, p-coumaroylquinic, protocatechuic and tannic acids, esters of caffeic and ferulic acids with quinic acid, flavonoids, catechin, epicatechin, rutin, trigonelline, caffeine, and quercetin, among others) is found in spent coffee grounds [10,11]. These bioactive compounds present numerous biological properties, including potent antioxidant, anticarcinogenic, anti-allergic, anti-inflammatory, antimicrobial, and antitumor properties, as well as beneficial properties related to neuroprotection [12]. As a source of valuable compounds with potential applications in the pharmaceutical, cosmetic, and food industries, spent coffee grounds are an interesting example of waste valorization in the agri-food industry [8,11].

The literature has indicated that the antioxidant capacity of spent coffee grounds ranges from 74.57 to 172 µmol Trolox equivalents/g dry spent coffee grounds [13,14], and the total phenolic content is between 20 and 30 mg gallic acid equivalents (GAE)/g dry spent coffee grounds [15]. Furthermore, there is chlorogenic acid (CGA) and its derivates, which are the result of the esterification of quinic acid and a hydroxycinnamic acid (e.g., caffeic, ferulic, or p-coumaric acids) and are a major family of phenolic compounds found in spent coffee grounds [16]. The structures of some CGAs commonly found in coffee are presented in Figure 1, as well as their trivial names. The results of the study of Okur et al. [17] are in accordance with this affirmation and revealed that CGA was the main phenolic compound found in spent coffee grounds. Hence, the concentration of CGAs present in spent coffee grounds can be four to seven times higher than their corresponding content in coffee brews [13]. Spent coffee grounds also present high amounts of methylxanthines, with caffeine being the most abundant compound recovered, constituting 1–2% of the spent coffee grounds’ dry weight. Both caffeine and CGA present remarkable health benefits related to their biological properties, including strong antioxidant capacity, providing protection against free radical damage and oxidative stress [18,19].

Coffee’s chemical composition can be strongly influenced by several parameters (e.g., coffee species, geographical origin, processing, etc.). Moreover, it has been demonstrated in the literature that coffees from different geographical origins present different amounts of phenolic compounds, even when belonging to the same species [20,21]. For instance, a study performed by Baeza et al. [21] determined that Ethiopian Arabica coffee presents a higher content of polyphenols than Colombian and Brazilian Arabica coffees. Another study carried out by Catelani et al. [20] showed that Timorese Arabica coffee had a superior concentration of polyphenols when compared to Brazilian Arabica coffee. Additionally, Zhu et al. [22] investigated the chemical composition of coffees from different geographical origins and observed that the samples presented decreasing amounts of phenolic compounds in the following order of origin: Ethiopia, Guatemala, Colombia, and Brazil. These differences in the chemical composition, being in the type or concentration of the chemical compounds, are also evident in the spent coffee grounds [23].

On the above basis, spent coffee grounds can be a rich source of bioactive compounds of interest to the pharmaceutical, cosmetic, and food industries. The extraction of bioactive compounds from spent coffee grounds has been reported in the literature, with researchers applying solid/liquid extraction while studying the effects of the main extraction parameters (e.g., time, temperature, and type of solvent) [16,17]. Other studies report the use of more conventional techniques to recover bioactive compounds from spent coffee grounds, such as Soxhlet extraction and gravity filtration. However, new alternative techniques have been proposed to minimize energy consumption and processing costs, as well as substitute toxic solvents with environmentally friendly ones and therefore reduce the negative impact on the environment, including supercritical fluids [24], microwaves [25], ultrasound [14], and high pressure [26]. Hence, the present work aimed at evaluating the antioxidant potential through a miniaturized quick, easy, cheap, effective, rugged, and safe (µ-QuEChERS) extraction combined with spectrophotometric techniques (UV-Vis), in addition to the identification and quantification of bioactive compounds via a miniaturized solid-phase extraction (µ-SPEed) coupled to ultrahigh performance liquid chromatography with photodiode array detection (UHPLC-PDA) in the spent coffee grounds from different geographical origins (Guatemala, Colombia, Brazil, Timor, and Ethiopia). The bioactive compounds identified in spent coffee grounds could be used in the food industry for the development of new food ingredients or products for human consumption, pharmaceutical and cosmetic formulations, like as antioxidant, antimicrobial, and anti-inflammatory agents, promoting their integral valorization within the circular bioeconomy concept.

## 2. Results and Discussion

### 2.1. Evaluation of the Bioactive and Antioxidant Potential of Spent Coffee Grounds

To evaluate the bioactive and antioxidant potential of the spent coffee grounds, a µ-QuEChERS extraction procedure was performed, and the extracts were analyzed through spectrophotometric assays (TPC, DPPH, and ABTS). The µ-QuEChERS procedure, firstly proposed by Anastassiades et al. [27] for the quantification of multi-residue pesticides in fruits and vegetables, is a simple, effective, and inexpensive way to extract and clean residual analytes from a variety of sample matrices. Recently, this methodology has been shown to be adequate for the analysis of polyphenols and other bioactive compounds in foods [28] and therefore was chosen for the extraction of bioactive compounds from the spent coffee grounds analyzed in the present work.

Phenolic compounds are recognized for having high antioxidant activity due to their redox properties, playing important roles in absorbing and neutralizing free radicals [29]. In the present work, the TPC was assessed using the Folin–Ciocalteu method, which is a colorimetric assay based on electron transfer reactions between the Folin–Ciocalteu reagent and phenolic compounds [30]. The TPC results for the spent coffee grounds from different geographical origins can be found in Figure 2. The values obtained for TPC ranged from 41.6 ± 2.1 to 53.7 ± 3.1 mg GAE/100 g DW, for Colombia and Brazil spent coffee grounds, respectively. No statistically significant differences (*p* < 0.05) were found among the samples from Guatemala, Brazil, Timor, and Ethiopia. However, the TPC obtained for Colombia was significantly different from all other samples except for Ethiopia. Additionally, the TPC obtained for the spent coffee grounds analyzed was considerably lower than the results reported in the literature. For instance, Mussatto et al. [31] were able to extract 18 mg GAE/g DW, while Panusa et al. [16] obtained extracts with TPC up to 36 mg GAE/g DW. The differences observed between the TPC values obtained for the spent coffee grounds samples could be related to the edaphoclimatic conditions (e.g., temperature, humidity, altitude, and geographical location) that influence the quantities in which these secondary metabolites occur in the coffee plants [32,33].

The antioxidant potential of the spent coffee grounds was determined using the DPPH and ABTS assays. Since these two assays use model synthetic radicals which are not directly associated with food or biological systems, frequent objections are raised; however, they are commonly applied due to their simplicity, acceptable repeatability, and low cost. Moreover, the literature demonstrates that DPPH and ABTS are the most preferably employed synthetic radicals in antioxidant assays [34,35,36]. The scavenging of the DPPH, which is a highly colored and stable free radical, is the basis of the popular DPPH antioxidant assay [37]. The values obtained for the DPPH scavenging assay, presented in Figure 3, differed from 50.6 ± 5.3 (Guatemala) to 78.1 ± 7.3 mg TE/100 g DW (Brazil). There were no statistically significant differences among the samples from Guatemala, Colombia, and Timor, while the value obtained for Ethiopia was only significantly different from Guatemala. Brazillian spent coffee grounds presented antioxidant capacity significantly superior to all samples, except for Ethiopia.

Additionally, an ABTS assay was performed to measure the antioxidant potential of the spent coffee grounds. The green–blue stable radical cationic chromophore ABTS^•+^ is generated by the reaction of the ABTS salt with a strong oxidizing agent (in this case, potassium persulfate). The ABTS assay is based on the suppression of its characteristic long-wave absorption spectrum caused by the reduction of the ABTS radical by hydrogen-donating antioxidants [38]. The values obtained for the scavenging of the ABTS radical (Figure 4) ranged between 0.5 ± 0.04 (Guatemala) and 1.8 ± 0.2 (Ethiopia) mg TE/100 g DW. The statistical analysis showed that the ABTS scavenging capacity varied significantly between all the samples, except for Colombia and Brazil. The values observed for the ABTS assay were considerably lower than those obtained in the DPPH assay for all samples. This could be related to the specific antioxidant capacity of the compounds present in the extracts combined with the free radicals used. Nenadis et al. [38] compared the order of antioxidant activity of specific phenolic compounds against the DPPH and ABTS radicals. The results showed that caffeic and chlorogenic acids, two phenolic compounds typically found in coffee, presented higher scavenging capacity against DPPH than ABTS.

A correlation between the TPC values and the antioxidant capacity should be expected since phenolic compounds are potent antioxidants, meaning that higher TPC corresponds to superior antioxidant capacity. However, this was not observed in this work’s results. This could be explained by the presence of non-phenolic compounds in the extracts analyzed that possess antioxidant capacity (for instance, caffeine) and aid in the reduction of free radicals.

### 2.2. Analysis of Bioactive Compounds by UHPLC-PDA

#### Validation of the µ-SPEed/UHPLC-PDA Methodology

The performance of the µ-SPEed/UHPLC-PDA method applied was validated in terms of selectivity, linearity, limit of detection (LOD), limit of quantification (LOQ), precision, and trueness (expressed as recovery %) to demonstrate that the method is suitable for the quantification of bioactive compounds in spent coffee grounds.

The selectivity of the method was assessed through the comparison of the chromatograms and PDA spectra obtained for the spent coffee grounds UHPLC-PDA analysis with those of analytical standards. The standards used for the identification were 3-CQA, 5-CQA, caffeine, caffeic acid, 4,5-diCQA, 1,5-diCQA, and 3,4-diCQA. The chromatograms obtained from the individual standards used in the identification of bioactive compounds are shown in Figure 5. There were not considerable interferences in the retention times at which the analytes of interest appear, proving that the method is selective and allowing the identification of these analytes in the different spent coffee grounds.

Furthermore, the capacity of the method to produce results directly proportional to the concentration of the analytes was evaluated. The calibration curves were obtained by performing the analytical approach with seven different concentrations of a mix solution of 5-CQA, caffeine, and caffeic acid, ranging from 0.5 to 75 µg/mL. The analytes 3-CQA, 4,5-diCQA, 1,5-diCQA, and 3,4-diCQA were quantified in 5-CQA equivalents. The results of the linearity evaluation are presented in Table 1. Linearity was observed for all the compounds within the concentration range utilized, with the coefficients of correlation (R^2^) higher than 0.998, proving a satisfactory adjustment of the obtained value to the calibration curve.

The LOD and LOQ values correspond to the lowest concentrations at which the analytes can be detected and quantified in the sample, respectively. The LOD and LOQ values calculated for each bioactive compound are shown in Table 1. The LODs observed were low, ranging between 10.3 (5-CQA) and 31.2 ng/mL (caffeic acid). Regarding the LOQs, the values obtained ranged from 32.4 (5-CQA) and 103 ng/mL (caffeic acid). Similar results were observed in previous works [39], indicating that the method utilized is acceptable for the quantification of small amounts of these bioactive compounds in spent coffee grounds.

The precision and accuracy of the µSPEed/UHPLC-PDA method used were also assessed by spiking one of the spent coffee grounds samples (Timor). The sample was spiked in three different concentrations of 0.2 µg/mL (low level, LL), 25 µg/mL (medium level, ML), and 75 µg/mL (high level, HL). The results for the precision and accuracy of the method are presented in Table 2. Precision was evaluated to determine the ability of the method to generate reproducible results. The parameters evaluated were the repeatability (intra-day), in which the analyses were performed on the same day repeatably, and intermediate precision (inter-day), in which the analyses were performed on different, non-consecutive days. The precision was expressed in the percentage of relative standard deviation (RSD%). The values obtained for the repeatability and intermediate precision were all below 16%, indicating the good performance of the method regarding these parameters. 

Finally, the accuracy of the method was assessed to determine the degree of agreement between the reference value (concentration of standard analyte spiked) and the actual value obtained after the analysis. The accuracy results were expressed in the percentage of recovery (Rec%). The closer to 100% the value obtained is, the more accurate the method is. Typically, the acceptability limits for the accuracy of an analytical method are set at ±25% [40]. The accuracy results obtained for the analytes used in this work (Table 2) ranged from 94.5 to 120 %. Since these values are within the acceptable limits defined, it was possible to conclude that the µSPEed/UHPLC-PDA methodology used can provide accurate results.

Overall, the validation results suggest that the µSPEed/UHPLC-PDA method is adequate for the extraction and quantification of bioactive compounds in spent coffee grounds.

### 2.3. Application of the µSPEed/UHPLC-PDA Method in the Spent Coffee Grounds Samples

After the µSPEed/UHPLC-PDA method was proved to be suitable for the determination of the target analytes, it was performed on the spent coffee grounds from different geographical origins. The results for the concentration of each bioactive compound found in the samples are shown in Table 3. The chromatographic areas for the bioactive compounds were collected according to the maximum wavelength of each compound’s PDA spectra: 273 nm for caffeine and 326 nm for 3-CQA, 5-CQA, caffeic acid, 4,5-diCQA, 1,5-diCQA, and 3,4-diCQA. The results were expressed in mg of the target analyte per 100 g of DW (mg/100 g DW).

The bioactive profile of the different spent coffee grounds analyzed are similar to one another, only differing in the concentrations in which they appear. Caffeine and 5-CQA were the two most abundant bioactive compounds found in all of the samples. Particularly, caffeine was the most concentrated compound in the spent coffee grounds from Guatemala (391.9 ± 27.3 mg/100 g DW), Colombia (386.8 ± 25.6 mg/100 g DW), Brazil (194.1 ± 4.8 mg/100 g DW), and Timor (381.1 ± 9.3 mg/100 g DW), while the sample from Ethiopia was richer in 5-CQA (247.7 ± 11.8 mg/100 g DW). After caffeine and 5-CQA, 3-CQA was the third compound found in higher concentration in all the samples, ranging from 8.3 ± 1.5 mg/100 g DW (Brazil) to 104.1 ± 20.7 mg/100 g DW (Guatemala). The other bioactive compounds analyzed were present in lower concentrations in the spent coffee grounds. The amount of 4,5-CQA ranged from 3.6 ± 0.3 to 11.0 ± 1.1 mg/100 g DW in Brazil and Guatemala spent coffee grounds, respectively, while 3,4-CQA was found in amounts between 0.6 ± 0.06 mg/100 g DW (Brazil) and 7.0 ± 0.7 mg/100 g DW (Guatemala). Finally, caffeic acid and 1,5-CQA were only quantified in the sample from Guatemala, as the other samples presented concentrations of these bioactive compounds below the LODs. Guatemalan spent coffee grounds presented higher concentrations of all the compounds analyzed in this work, whilst spent coffee grounds from Brazil presented lower abundance. Among the CGAs identified in spent coffee grounds, only CQAs were found and 5-CQA was the most abundant compound quantified. This outcome was expected, as about 80% of the total CGA content corresponds to CQAs in green coffee, and 5-CQA corresponds to almost 60% of the total CGA content [41]. Generally, monocaffeoylquinic acids (e.g., 3-CQA and 5-CQA) occur in higher concentrations in coffee when compared to dicaffeoylquinic acids (e.g., 4,5-diCQA, 1,5-diCQA, and 3,4-diCQA), with the latter examples appearing in such low concentration that they cannot be quantified or even detected [42]. Similar results were obtained in the present work, with some differences observed between the values obtained from the different samples, probably due to the samples being from different geographical origins.

Similar to the results obtained in this work, relatively high amounts of caffeine and CQAs have been previously observed in spent coffee grounds. However, the literature reports the recovery of higher concentrations of these bioactive compounds from different types of spent coffee grounds. For instance, Panusa et al. [16] were able to recover as much as 6.1 mg of CQAs/g DW and 11.5 mg of caffeine/g DW from spent coffee grounds of a blend of 40% Arabica and 60% Robusta; however, in the same work, they report the recuperation of about 2.3 mg of CQAs/g DW and 1 mg of caffeine/g DW from 100% Arabica spent coffee grounds, which is less than what was obtained in the present work. Additionally, Shang et al. [26] analyzed several spent coffee samples from the Arabica variety and one Robusta spent coffee grounds and observed that the different Arabica samples presented caffeine concentrations ranging from 3.19 to 7.45 mg/g DW and 5-CQA concentrations between 51.72 and 213.98 mg/g DW. The differences observed in the concentrations at which these bioactive compounds are found in the spent coffee grounds samples are probably related to the edaphoclimatic factors that influence the concentrations at which these secondary metabolites are present [32,33]. Shang et al. [26] also observed that Robusta presented the highest content of caffeine. This is due to the Robusta variety being richer in secondary metabolites than Arabica coffee, in particular, its caffeine content is usually reported as twice the concentration of caffeine found in Arabica coffee [43].

Caffeic acid was only found in concentrations superior to the LOD value in the spent coffee grounds from Guatemala, where its content was determined to be 3.655 ± 0.689 mg/100 g DW. This result is below the caffeic acid content reported in the literature on spent coffee grounds (9.36 mg/100 g of oil extracted from spent coffee grounds) [44]. Caffeic acid is a hydroxycinnamic acid generally found in coffee esterified with quinic acid to form chlorogenic acids [45]. Although a part of the chlorogenic acid is degraded during coffee roasting, originating caffeic acid, it can undergo further reactions to form aromatic compounds [46]. For these reasons, the content of caffeic acid can depend on several factors, including coffee variety and roasting conditions [47].

### 2.4. Possible Applications for the Bioactive Compounds Identified in the Spent Coffee Grounds Samples

As the results demonstrate, spent coffee grounds can be a rich source of some bioactive compounds. Depending on the extraction method used in the preparation of the coffee beverage, great amounts of these compounds remain in the coffee grounds matrix and are usually thrown away as waste. In this section, some possible applications for the bioactive compounds identified in the spent coffee grounds analyzed (e.g., caffeine, chlorogenic acids, and caffeic acid) will be discussed.

Caffeine is one of the most abundant bioactive compounds in coffee, and great amounts of it remain in spent coffee grounds after the beverage’s preparation. Caffeine is the most popularly used psychostimulant worldwide [48] and has been proven to have several biological effects on the human body, including its role in affecting the pathophysiology of neurodegenerative disorders (e.g., Alzheimer’s and Parkinson’s Diseases) and cardiovascular diseases, in the treatment of neonatal apnea, and as an analgesic adjuvant [49,50]. These properties are highly related to its rapid absorption by the gastrointestinal tract [48]. Due to the benefic effects attributed to caffeine, it can have an array of applications in the pharmaceutical, cosmetic, biocide, and food industries. In pharmaceutics, caffeine is used as a respiratory stimulant, as an additive in drugs to improve the analgesic effect, and as a regulator of appetite [51]. In the cosmetic field, the application of caffeine in topical cosmetic products proved to prevent excessive fat accumulations in the skin, promote lymphatic drainage, and protect the skin from photodamage [52]. Caffeine has been highly reported to enhance the beneficial properties of dermatologic products due to its antioxidant and phosphodiesterase inhibitory effects, as well as being a cosmetic and nutraceutical ingredient [53]. Another interesting application of caffeine that has been reported is as a biocide, specifically against decaying fungi and termites in wood protection [54], and it has proved to be an eco-friendly alternative to the artificial biocidal agent. Additionally, Hollingsworth et al. [55] showed that caffeine can be a potential molluscicide to repel slugs and snails from food crops in agriculture.

CGAs are the main class of bioactive compounds in coffee, and identically to caffeine, copious sums of them remain in spent coffee grounds. The effects of CGAs on human health derive mainly from their antioxidant, anti-inflammatory, anticancer, and hepatoprotective properties, among other properties. These characteristics are mainly attributed to 5-CQA since it is the most abundant of its class present in coffee [56,57]. CGAs can also be applied in several fields in the industry, such as in the pharmaceutical, cosmetic, and food industries. The topical application of CGAs was proved to have an inhibitory effect on tumor promotion in mouse skin [58] and to accelerate the process of excision wound healing in rats [59]. Moreover, Shao et al. [60] proposed the encapsulation of CGA in β-cyclodextrin and the potential application of the complexes in the food industry as enhanced antioxidants and co-pigments. Other authors reported the creation of chitosan grafted with chlorogenic acid to be used as a potential preservative agent and edible coating material in peach fruit [61].

Caffeic acid is a naturally occurring hydroxycinnamic acid produced as a secondary metabolite of the shikimate way. This bioactive compound is widely recognized for its antioxidant and anticoagulant activity [62]. Although caffeic only appeared in concentrations above the LOD in the spent coffee grounds from Guatemala, it is a potent antioxidant, and some applications have been studied; Hallan et al. [63] encapsulated caffeic acid within solid-lipid nanoparticles and ethosomes to improve the therapeutic potential.

Overall, the compounds identified and quantified in the spent coffee grounds analyzed present several bioactivities. However, the amounts quantified in the spent coffee grounds are well below the concentrations that can be found in ground coffee. The caffeine content in ground coffee can be as high as 6 g/100 g of DW (for the Arabica variety) [64], which is approximately twenty times the value obtained in this work. This makes spent coffee grounds a less viable source of these compounds since higher yields can be obtained from other raw materials (e.g., coffee beans and black tea leaves). Nevertheless, the recovery of value-added compounds from waste materials is highly prioritized, in light of the biorefinery framework. This causes spent coffee grounds to be an interesting source of bioactive compounds and an important contribution to the circular economy.

## 3. Materials and Methods

### 3.1. Reagents and Standards

The HPLC grade acetonitrile (ACN, CH_3_CN) and methanol (MeOH, CH_3_OH) were acquired from Fisher Scientific (Loughborough, UK), whereas the 3-CQA, 1,5-CQA, 3,4-CQA, and 4,5-CQA (>98.0%) were purchased from Biopurify Phytochemicals LTD (Chengdu, China). The Folin–Ciocalteu solution, 2,2-diphenyl-1-picrylhydrazyl, (DPPH, C_18_H_12_N_5_O_6_), 6-hydroxy-2,5,7,8-tetramethylchromane-2-carboxylic acid (Trolox, C_14_H_18_O_4_, 98.0%), gallic acid (GA, C_7_H_6_O_5_, 98.0%), and caffeic acid (C_9_H_8_O_4_, 99.0%) were supplied by Fluka (Munich, Germany). Potassium persulfate (K_2_S_2_O_8_, 99.0%) and sodium nitrite (NaNO_2_) were purchased from Merck^®^ (Buchs, Switzerland); disodium phosphate dihydrate (Na_2_HPO_4_·2H_2_O, 99.0%), potassium dihydrogen phosphate (KH_2_PO_4_, 99.0%), trisodium citrate dihydrate (C6H9Na3O9, 99.0%), and formic acid (FA, CH_2_O_2_, 98.0%) were acquired from Panreac Applichem (Barcelona, Spain). Aluminum chloride (AlCl_3_), potassium chloride (KCl, 99.5%), and ethyl acetate (EtAc, C_4_H_8_O_2_, 99.7%) were supplied by Riedel-de Haën^®^ (Seelze, Germany). The 2,29-azinobis-(3-ethylbenzothiazoline-6-sulfonic acid) radical cation (ABTS, C_18_H_18_N_4_O_6_S_4_, 98.0%), sodium hydrogencitrate sesquihydrate (C_6_H_9_NaO_8_, 99.0%), and 5-CQA (C_16_H_18_O_9_, ≥95.0%) were obtained from Sigma-Aldrich (Buchs, Switzerland), and sodium hydroxide (NaOH, 98.0%) was purchased from Eka Chemicals AB (Amsterdam, The Netherlands). Sodium carbonate (Na_2_CO_3_, 99.7%) was supplied by Labsolve^®^ (Lisboa, Portugal). The 2 mL DisQuETM dSPE tubes containing the sorbents (150 mg of MgSO_4_ and 25 mg of PSA) used in the µ-QuEChERS clean-up step were obtained from Waters (Milford, MA, USA). Ultrapure water (H_2_O) (18 MΩ cm) was obtained from a Milli-Q water purification system (Millipore, Burlington, MA, USA).

### 3.2. Sample Preparation

The roasted coffee samples were purchased from the Awaked Company (Caldas da Rainha, Portugal). The samples acquired were composed of 100% Arabica coffee and were single-origin coffees from Guatemala, Colombia, Brazil, Timor, and Ethiopia. The five roasted coffee samples were ground until a thin powder was obtained. After that, 6 g of each sample was used to make 20 mL of expresso in a coffee machine, and the spent coffee grounds were collected and stored in a flask for posterior analysis.

### 3.3. Evaluation of the Bioactive and Antioxidant Potential of Spent Coffee Grounds

#### 3.3.1. µ-QuEChERS Extraction

The µ-QuEChERS extraction was performed according to the procedure optimized by Casado et al. [65]. Briefly, 0.5 g of the lyophilized spent coffee grounds were added to 0.4 g of the µ-QuEChERS mixture (buffered salts) in the following proportion 4:1:1:0.5 of MgSO_4_, NaCl, C_6_H_9_Na_3_O_9_, C_6_H_9_NaO_8_, respectively. Then, 2 mL of a solution of ACN:EtAc (1:1, *v*/*v*), containing 0.1% of FA was added and the flask was vortexed for 10 s. The mixture was then submitted to ultrasonic agitation for 5 min in an ultrasonic bath and centrifuged for 5 min at 5000 rpm. After this, the supernatant (~1400 µL) was transferred to a 2 mL DisQueTM dSPE clean-up tube containing 150 mg of MgSO_4_ and 25 mg of PSA. The tube was vortexed for 30 s and centrifuged for 5 min at 4000 rpm. The extract was, finally, filtered through 0.22 µm PTFE syringe filters (BGB Analytik, VA, USA) into a vial, and stored at −20 °C until the analysis.

#### 3.3.2. Total Phenolic Content

The total phenolic content (TPC) of the spent coffee grounds was determined using the Folin–Ciocalteu procedure, as described by Figueira et al. [66], with a few modifications. Briefly, the extracts obtained by the µ-QuEChERS procedure were diluted in water up to a 3 mL final volume. Then, 300 µL of Folin–Ciocalteu solution was added to the reaction tube, followed by 1200 µL of 20% (*w*/*v*) Na_2_CO_3_ solution and 1500 µL of H_2_O. The mixture was homogenized and incubated for 30 min in the dark and at room temperature (25 ± 1 °C). After the incubation time, the absorbance was measured using a UV-Vis spectrophotometer (Lambda 25, Perkin Elmer, Waltham, MA, USA) at λ = 765 nm. The results were expressed in mg of GA equivalents (GAE)/100 g DW (dry weight) after the absorbance registered was interpolated in the calibration curve (y = 0.052x + 0.008, where y is the absorbance and x the concentration, R^2^ = 0.992) prepared with standard solutions with different concentrations of GA (from 0.5 to 15 µg/mL). The TPC was assessed in triplicate for each sample.

#### 3.3.3. 2,2-Diphenyl-1-picrylhydrazyl Scavenging Assay

The DPPH assay was performed according to Woraratphoka et al. [34] to measure the free radical properties of the spent coffee grounds. A DPPH stock solution was prepared by dissolving 24 mg of DPPH in 100 mL of MeOH. Before the reaction, the stock solution was diluted to obtain a working solution with an absorbance of ~0.9, and the absorbance value was registered. Then, the µ-QuEChERS extracts were diluted to a final volume of 100 µL and added to a reaction tube containing 3.9 mL of the DPPH working. The mixture was homogenized and incubated for 45 min in the dark and at room temperature. The absorbance was measured at λ = 515 nm using a UV-Vis spectrophotometer. The free radical scavenging capacity (AAR(DPPH)) against DPPH was calculated using the following formula (calibration curve): ln(%ΔA515) = 0.763 × ln(AAR(DPPH)) − 0.077 (R^2^ = 0.985), where %ΔA515 = [(A515(0) − A515(45))/A515(0)] × 100, where A515(0) is the absorbance value measured at the beginning of the reaction and A515(45) is the absorbance value measured after 45 min of reaction. The calibration curve was obtained by performing the reaction procedure using standard solutions with different concentrations of Trolox (from 5 to 400 µg/mL), and the results were expressed in mg Trolox equivalents (TE)/100 g of DW. The antioxidant capacity against DPPH was assessed in triplicate for each sample.

##### 3.3.4. 2,29-Azinobis-(3-ethylbenzothiazoline-6-sulfonic Acid) Assay

The ABTS assay was adapted from the procedure reported by Paixão et al. [67] to determine the antioxidant capacity of spent coffee grounds against the stable ABTS^+^ radical cation. Briefly, a stock solution of ABTS (20 mM) was prepared in 50 mL of phosphate-buffered saline (PBS, pH 7.4) and 200 µL of 70 mM potassium persulfate solution was added. The solution was stored in the dark at room temperature (25 ± 1 °C) for 16 h. The ABTS solution was diluted with PBS until an absorbance value of ~0.9 was obtained, and the absorbance of the working solution was registered. Then, 12 μL of the µ-QuEChERS extracts were added to 3 mL of the diluted ABTS solution. The mixture was then homogenized and incubated for 20 min in the dark and at room temperature. After the incubation, the absorbance was measured at 734 nm using a UV-Vis spectrophotometer. The free radical scavenging capacity against ABTS (AAR(ABTS)) was calculated using the following formula (calibration curve): I = 0.045 × AAR(ABTS) + 0.709 (R^2^ = 0.991), where I = [(AB − AA)/AB] × 100, where I is the percentage of inhibition of ABTS*, AB the absorbance of a blank sample (t = 0 min), and AA is the absorbance after 20 min of adding the extracts. The calibration curve was obtained by performing the reaction procedure using standard solutions with different concentrations of Trolox (from 10 to 600 µg/mL), and the results were expressed in mg Trolox equivalents (TE)/100 g of DW. The antioxidant capacity against ABTS was assessed in triplicate for each sample.

### 3.4. Analysis of Bioactive Compounds by UHPLC-PDA

#### 3.4.1. Sample Preparation

All spent coffee grounds were lyophilized before the extraction of bioactive compounds. The samples were then subjected to maceration to obtain a liquid extract, in which 100 mL of distilled water was added to 1.5 g of each sample and left under magnetic stirring for 1 h in a bath at 50 °C. After maceration, the sample was centrifuged, and the supernatants were transferred to a flask and stored in the cold until the extraction was conducted.

#### 3.4.2. Extraction of Bioactive Compounds via µ-SPEed

The extraction of the bioactive compounds from the spent coffee grounds was obtained using µ-SPEed^®^. The extraction conditions applied were adapted from the procedure optimized by Casado et al. [68]. Before the extraction, the sample pH was adjusted to pH 2 and all samples were filtered through 0.22 µm PTFE syringe filters (BGB Analytik, VA, USA). A programmable digital syringe driver, digiVOL^®^, equipped with a 250 µL syringe (ePrep^®^, Oakleigh, Australia), was used for the µ-SPEed^®^ experiments. The sorbent used was a porous polystyrene-dininylbenzene reversed phase (PS/DVB-RP). For each extraction, the sorbent was activated with 100 µL of MeOH and conditioned with 100 µL of acidified H_2_O (0.1% FA). After this, the sample was passed through the sorbent 10 times. No washing step was applied, and the analytes were directly eluted with 50 µL of MeOH:H_2_O (95:5, *v*/*v*) containing 0.1% FA into the vial for chromatographic analysis. After each extraction, the sorbent was reconditioned by passing 2 × 100 µL of MeOH and 100 µL of acidified H_2_O (0.1% FA) to prevent the carry-over of analytes to the next extraction. The extraction was performed in duplicate for each sample.

#### 3.4.3. UHPLC-PDA Analysis

The chromatographic analysis of the spent coffee grounds was performed using a UHPLC system (Waters Ultra-High Performance Liquid Chromatography Acquity H-Class system) (Milford, MA, USA) equipped with a quaternary solvent manager (QSM), an Acquity sample manager (SM), a column heater, a degassing system, and a photodiode array (2996 PDA) detector. The column used for the separation of the analytes was an Acquity HSS T3 analytical column (2.1 mm × 50 mm, 1.7 µm particle size) packed with a trifunctional C18 alkyl phase (Waters, Milford, MA, USA). The chromatographic separation of the target analytes was achieved with the column at 40 °C, using a mobile phase composed of acidified H_2_O (0.1% FA) (solvent A) and ACN (solvent B). The gradient condition was varied as follows: (a) initially 85% A and 15% B, (b) 7 min 63% A and 37% B, and (c) 8 min 85% A and 15% B at a constant flow rate of 0.250 mL/min. Each analysis was followed by a 2 min re-equilibration time before the next injection. The total analysis time was 10 min. The injection volume was 2 µL and the sample manager compartment was kept at 20 °C. The PDA data was registered at 273 and 326 nm, according to the maximum wavelength of the analyzed compounds. The Empower software 2.0 (Waters, Milford, MA, USA) was used to drive the whole UHPLC configuration and for data collection. The identification of the target analytes was achieved by comparing the retention times and UV spectrum with those obtained for pure standards using the same instrumental conditions. Each extract was analyzed in triplicate.

#### 3.4.4. Validation of the µ-SPEed/UHPLC-PDA Methodology

The analytical performance of the proposed methodology (µ-SPEed/UHPLC-PDA) was validated in terms of selectivity, linearity, limit of detection (LOD), limit of quantification (LOQ), precision (intra-day and inter-day, expressed as the percentage of relative standard deviation, %RSD), and accuracy (expressed as the percentage of recovery, %Rec.), to assure that the method is suitable for the determination of the target analytes in the spent coffee grounds samples.

The selectivity of an analytical method corresponds to the extent to which it can determine analytes in a complex mixture without interference from other components in the mixture. The selectivity was assessed by analyzing the spent coffee grounds using the µ-SPEed/UHPLC-PDA method and comparing them with the standard solutions. The absence of interferences in the retention time and wavelength of the target analytes proves that the proposed methodology is selective.

The linearity of a method is the capacity of an analytical method of producing results directly proportional to the concentration of an analyte within a range of concentration. It can be assessed by a measure of how well a calibration plot of response (usually chromatographic area of the peak) vs. the concentration approximates a straight line. The linearity of the µ-SPEed/UHPLC-PDA method was determined by preparing a calibration curve with seven points (*n* = 7) with concentration ranging from 0.2 to 75 µg/mL.

The LOD is the lowest concentration of an analyte from which it is possible to deduce its presence in the sample. Similarly, the LOQ is the smallest concentration of an analyte that can be determined in the sample. The calculation of LOD and LOQ are similar, and one can be inferred from the other. The instrumental LOD can be determined by using a very low concentration of the target analytes and comparing its signal-to-noise (S/N) ratio. For LOD, the signal (chromatographic peak) should be at least three times superior to noise and should yield acceptable identity and purity. Regarding LOQ, the signal should be at least 10 times the noise. In the present work, the LOD and LOQ of the µ-SPEed/UHPLC-PDA method applied were determined considering the concentration that produced a signal-to-noise ratio equal or higher to three and ten, using the lowest standard concentration of the calibration curve.

Precision is a measure of the ability of the method to generate reproducible results. Precision can be considered on three levels: repeatability, intermediate precision, and interlaboratory precision (also known as reproducibility), and its evaluation should be performed on homogenous authentic samples or artificially prepared samples. In the present work, precision was assessed in terms of repeatability, by performing a series of repeated analyses within a short period of time (intra-day) and intermediate precision, by repeating the analyses on different, non-consecutive days (inter-day). One of the spent coffee grounds (Timor) was used for the evaluation of precision and was spiked in three different levels of concentration: low level (LL, 0.2 µg/mL), medium level (ML, 25 µg/mL), and high level (HL, 75 µg/mL). Intra-day precision was determined by analyzing six replicates in triplicate (*n* = 6) for each spiking level, while inter-day precision was determined by analyzing six replicates of each level daily through three different days in triplicate (*n* = 18). The precision was expressed in terms of the percentage of relative standard deviation (%RSD). The trueness (extraction efficiency) of the method, expressed as recovery percentage (%), was evaluated by spiking the spent coffee grounds (Timor) in triplicate using the same spiking levels used in the precision assays. The recovery values were calculated by comparing the areas obtained for spiked samples with those obtained for simulated samples (samples spiked at the same concentration levels but at the end of the extraction process), prior to their chromatographic analysis.

### 3.5. Statistical Analysis

Statistical analysis was carried out using MetaboAnalyst 5.0, which comprises the data pre-processing to eliminate metabolites with missing values (MV) and normalization (data transformation using data scaling by mean-center and cubic root). The normalized data was processed using the one-way ANOVA, followed by Tukey’s test for post hoc multiple comparisons of means to identify significant differences between the spent coffee grounds from different geographical origins.

## 4. Conclusions

In this work, the bioactive and antioxidant potential of spent coffee grounds from different geographical origins (Guatemala, Colombia, Brazil, Timor, and Ethiopia) were determined through a µ-QuEChERS method coupled to spectrophotometric techniques (total phenolic content (TPC), and DPPH and ABTS scavenging assays). The highest value for the TPC was obtained for the sample from Brazil (53.7 ± 3.1 mg GAE/100 g DW), while Colombia presented the lowest value (41.6 ± 2.1 mg GAE/100 g DW). As for the DPPH assay, the highest scavenging activity was presented by Brazil (78.1 ± 7.3 mg TE/100 g DW) and the lowest for Guatemala (50.6 ± 5.3 mg TE/100 g DW). In addition, the extracts presented higher scavenging capacity against the DPPH radical, compared to the ABTS. The values obtained could be related to the fact that caffeic and chlorogenic acids, two bioactive compounds typically found in coffee, presented higher scavenging capacity against DPPH than ABTS.

Finally, the identification of the bioactive compounds responsible for the antioxidant activity attributed to the spent coffee grounds was attempted using a sensitive and improved analytical method based on the µ-SPEed/UHPLC-PDA method. This method allowed the reduction of the amounts of sample and organic solvents, leading to an improved cost-effective, and environmentally friendly microextraction strategy, which meets the Green Analytical Chemistry principles. Moreover, the little requirement for sample preparation enhances the detection and quantification of the analytes means that this method also reduces the time and cost of the analysis. A satisfactory figure of merit was achieved for selectivity, linearity, LOD, LOQ, trueness, and intra- and inter-day precision, which demonstrates the feasibility and practicability of the method for quantifying compounds in spent coffee grounds from different geographical origins. Seven bioactive compounds, namely 3-CQA, 5-CQA, caffeine, caffeic acid, 4,5-diCQA, 1,5-diCQA, and 3,4-diCQ, were identified and quantified. Caffeine and 5-CQA were the most abundant bioactive compounds found in all the samples, and their highest concentrations were found in the sample from Guatemala (385.3 ± 24.4 and 391.9 ± 27.3 mg/100 g DW, respectively). These bioactive compounds have been recognized in the literature for their impressive biological effect, hence why they present several possible applications in the pharmaceutical, cosmetic, and food industries, among others. 

The results obtained in this work provide insight into the chemical composition of spent coffee grounds, allowing the identification and quantification of the bioactive compounds that remain in the matrix. Moreover, it was possible to conclude that this coffee industry residue can be a rich source of numerous compounds with industrial interest. However, more research must be done regarding the optimization of the extraction techniques used to recover as many of these compounds as possible. Furthermore, more studies need to be conducted to achieve an integrated cascade biorefinery approach to properly valorize spent coffee grounds. 

## Figures and Tables

**Figure 1 molecules-27-07504-f001:**
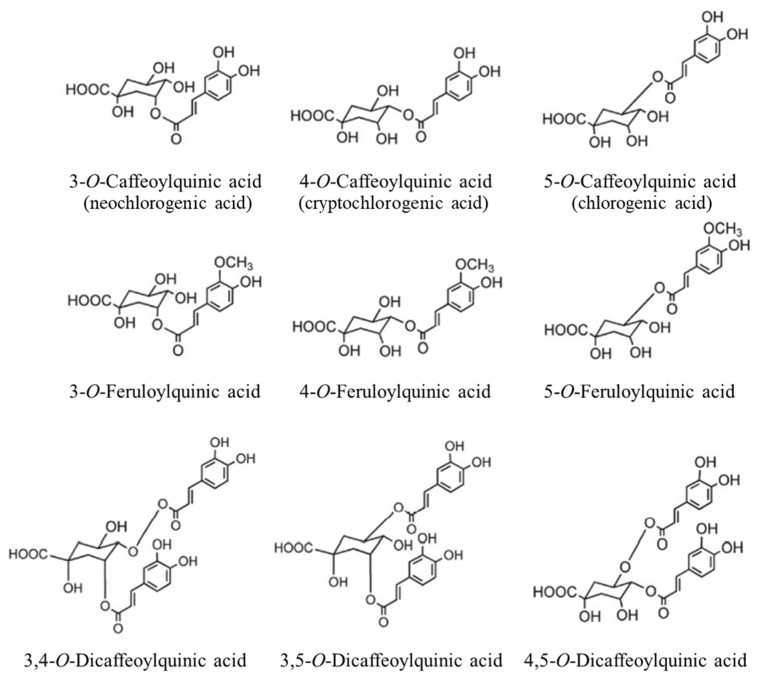
Molecular structure of some CGAs commonly found in coffee and their trivial names.

**Figure 2 molecules-27-07504-f002:**
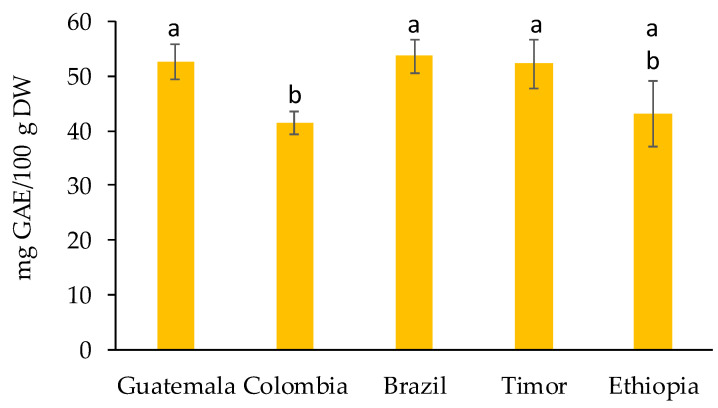
Total phenolic content of spent coffee grounds from different geographical origins expressed in mg GAE/100 g DW. Different superscript letters indicate significant differences (*p* < 0.05) among spent coffee grounds from different geographical origins.

**Figure 3 molecules-27-07504-f003:**
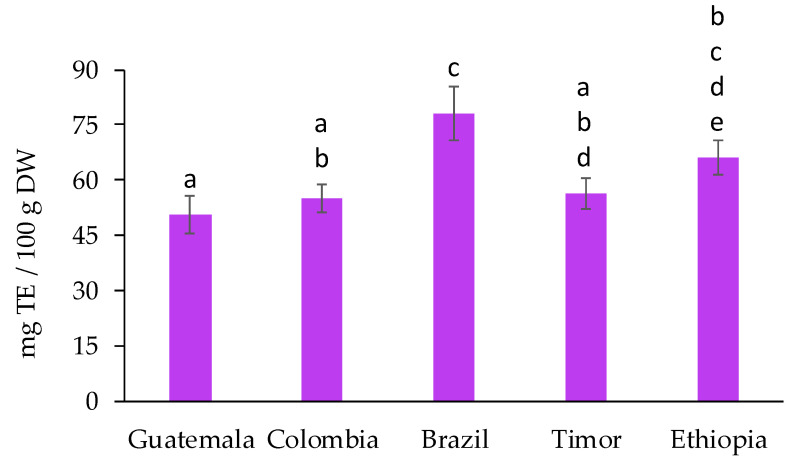
DPPH scavenging assay for spent coffee grounds from different geographical origins, expressed in mg of TE/100 g DW. Different superscript letters indicate significant differences (*p* < 0.05) among spent coffee grounds from different geographical origins.

**Figure 4 molecules-27-07504-f004:**
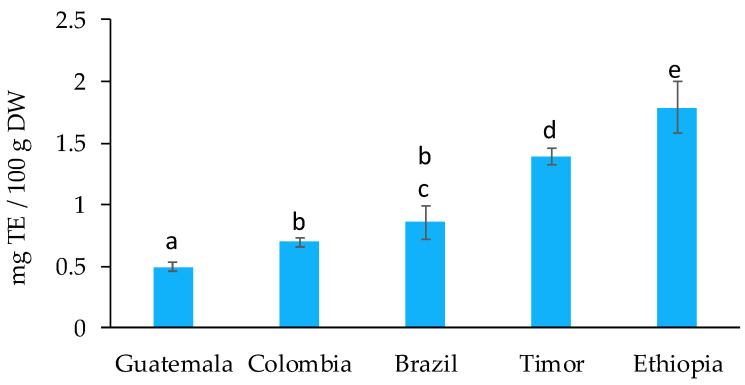
ABTS assay for spent coffee grounds from different geographical origins, expressed in mg of TE/100 g DW. Different superscript letters indicate significant differences (*p* < 0.05) among spent coffee grounds from different geographical origins.

**Figure 5 molecules-27-07504-f005:**
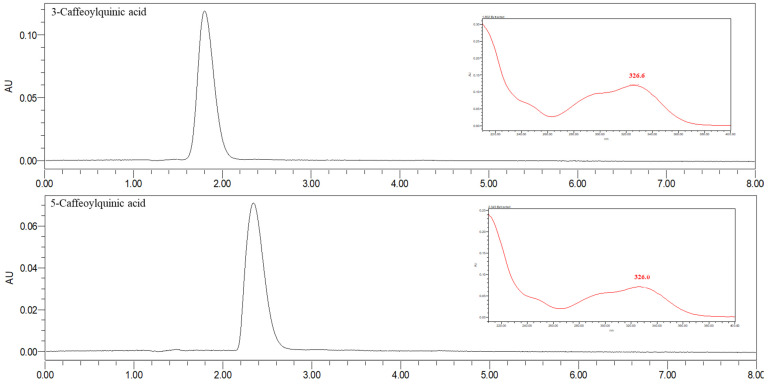
Chromatograms obtained for the individual standards and PDA spectra acquired for each peak used for the identification of the bioactive compounds in spent coffee grounds (acquired at 273 nm for caffeine and at 326 nm for the remaining analytes).

**Table 1 molecules-27-07504-t001:** Linearity (R^2^) evaluation and limits of detection and quantification through the µSPEed/UHPLC-PDA method.

RT (min)	Analyte	λ_max_ (nm)	Linear Range (µg/mL)	Calibration Curve		LOD (ng/mL)	LOQ (ng/mL)
2.45	5-CQA	326	0.2–75	Equation	y = 72508x + 57908	10.3	32.4
				R^2^	0.998		
2.85	Caffeine	273	0.2–75	Equation	y = 67275x − 23743	29.9	998.5
				R^2^	0.999		
3.94	Caffeic acid	326	0.2–75	Equation	y = 128542x + 57103	31.1	102.9
				R^2^	0.999		

RT—Retention time; 5-CQA—5-Caffeoylquinic acid; LOD—Limit of detection; LOQ—Limit of quantification.

**Table 2 molecules-27-07504-t002:** Precision and trueness of the µSPEed/UHPLC-PDA method.

Analyte	Spiking Level (µg/mL)	Precision (RSD%)		Recovery ± RSD
		Intra-Day	Inter-Day	
5-CQA	0.2	1.47	15.72	98.5 ± 9.5
	25	6.21	8.47	112 ± 2.7
	75	6.06	6.19	108 ± 2.6
Caffeine	0.2	4.68	15.64	94.5 ± 4.9
	25	2.90	12.94	101 ± 4.9
	75	2.69	6.88	120 ± 1.5
Caffeic acid	0.2	4.23	11.26	112 ± 5.9
	25	1.78	10.45	110 ± 7.3
	75	1.52	7.91	109 ± 8.0

RSD—percentage of relative standard deviation; 5-CQA—5-caffeoylquinic acid.

**Table 3 molecules-27-07504-t003:** Identification and quantification of bioactive compounds in spent coffee grounds from different geographical origins through µSPEed/UHPLC-PDA.

RT (min)	Analyte	λ_max_ (nm)	Concentration (mg/100 g DW) ± Standard Deviation				
			Guatemala	Colombia	Brazil	Timor	Ethiopia
1.83	3-CQA *	326	104.1 ± 20.7 ^a^	23.9 ± 4.0 ^b^	8.3 ± 1.5 ^c^	24.8 ± 2.2 ^b^	53.8 ± 5.30 ^e^
2.45	5-CQA	326	385.3 ± 24.4 ^a^	348.9 ± 26.3 ^a^	167.3 ± 10.5 ^b^	330.6 ± 29.6 ^a^	247.7 ± 11.8 ^c^
2.85	Caffeine	273	391.9 ± 27.3 ^a^	386.8 ± 25.6 ^a^	194.1 ± 4.8 ^b^	381.1 ± 9.3 ^a^	198.6 ± 6.8 ^c^
3.94	Caffeic acid	326	3.7 ± 0.7	<LOQ	<LOQ	<LOQ	<LOQ
5.41	4,5-diCQA *	326	11.0 ± 1.1 ^a^	8.8 ± 1.34 ^b,a^	3.6 ± 0.3 ^c^	7.5 ± 0.5 ^b^	4.5 ± 0.2 ^c^
5.71	1,5-diCQA *	326	2.3 ± 0.5	<LOQ	<LOQ	<LOQ	<LOQ
6.12	3,4-diCQA *	326	7.0 ± 0.7 ^a^	4.8 ± 0.7 ^b^	0.6 ± 0.06 ^c^	3.5 ± 0.3 ^d^	1.6 ± 0.2 ^e^

RT—retention time; <LOQ: below the limit of quantification; 3-CQA—3-Caffeoylquinic acid; 5-CQA—5-Caffeoylquinic acid; 4,5-diCQA—4,5-Dicaffeoylquinic acid; 1,5-diCQA—1,5-Dicaffeoylquinic acid; 3,4-diCQA—3,4-Dicaffeoylquinic acid. * 5-CQA equivalents. Different superscript letters indicate significant differences (*p* < 0.05) among spent coffee grounds from different geographical origins.

## Data Availability

All the data in this research are presented in the manuscript.

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
