# Peer review of "Bioactive Compounds and Antioxidant Activity from Spent Coffee Grounds as a Powerful Approach for Its Valorization"

_molecules, 2022, doi:10.3390/molecules27217504_

Round 1

Reviewer 1 Report

The manuscript presented by Andrade, Perestrelo and Câmara presents the results of analysis carried out with spent coffee material from five geographical origins (Guatemala, Colombia, Brazil, Timor, and Ethiopia). The work was directed to the identification by UHPLC-PDA, measurement of the total phenolic contents and bioactivity of compounds of industrial interest. The theme is of broad interest since coffee generates a huge ammount of residues and by-products that can be valored towards circular bioeconomy. However, the theme is also widely studied and the literaure has many works in this area.

Although references are ok, the authors missed important literature data from the current year.

In general, the manuscript is well prepared, but some aspects must be reviewed. See, for example, lines 167-170.

1. Introduction

Line 72. Please, correct: “… are a major phenolic compound..”

Since the work is a comparison among spent coffee grounds from different geographical origins, some data about the differences among the coffee from these regions should be presented. To evaluate the differences reported by the authors in this work, it is necessary to understand first if the species used are the same, as well as type (premium, traditional, etc). It is important to have these data in the introductory section

Line 183. Since Figure 4 cannot fit a single page, please, bring part of this figure to page 5, to remove the empty space.

2. Results and Discussion

Line 108. Please, add a reference and a brief explanation why µ - QuEChERS extraction procedure was chosen in this work.

The assays used (TPC, DPPH and ABTS) are very simple and, although highly used, they are not very specific and present several limitations, as the authors mention in the lines 133-134. To explain the choice of DPPH and ABTS assays, the authors say “...literature demonstrates that DPPH and ABTS are the most preferably em- 136 ployed synthetic radicals in antioxidant assays [28–30] …”. However, two of the references cited in this part (28 and 29) are from the group. In my opinion, these two references report the preference of the group by these assays, but do not represent the literature in general. Therefore, I recommend to replace these two references by references from authors which do not signg the current work.

Lines 141-144: Here and in some other places, difference among the samples was detected, but the authors do not discuss the possible reasons for the difference. This is what I was talking about earlier; it should be highly desirable to add some data about the type of grounds used in the works, since there may be some characteristics that could explain the results. Even literature data could be used trying to explain or to hypothesize the reasons for the differences.

Lines 204-206: Please, add a reference to confirm that the method is acceptable.

Line 252. There is not Figure 20.

Line 265-268. Please, verify if the characteristics of the different grounds can explain the fact that caffeic acid and 1,5-CQA were found only in specific samples. In the discussion (lines 281-285 and then in the following lines), there is inference about the varieties (Arabica and Robusta), and this is an example of the type of discussion/comparisons that are missing in the manuscript (introduction and discussion).

Lines 301 and followings. To have this section, it is necessary to give an ideia of the yield of the compounds in relation to commercial sources, because if the yield is very low in comparison with the current industrial sources, the use of spent coffee grounds is not competitive. I suggest to be clear if these residues are real potential sources of bioactive compounds; econnomically viable, I mean. In my opinion, it is not necessary to find high levels of caffeic acids and other compounds to have the work published, but I think that the results must be interpreted in order not to confuse the reader.

Author Response

The manuscript presented by Andrade, Perestrelo and Câmara presents the results of analysis carried out with spent coffee material from five geographical origins (Guatemala, Colombia, Brazil, Timor, and Ethiopia). The work was directed to the identification by UHPLC-PDA, measurement of the total phenolic contents and bioactivity of compounds of industrial interest. The theme is of broad interest since coffee generates a huge ammount of residues and by-products that can be valored towards circular bioeconomy. However, the theme is also widely studied and the literaure has many works in this area.

Although references are ok, the authors missed important literature data from the current year.

In general, the manuscript is well prepared, but some aspects must be reviewed. See, for example, lines 167-170.

  1. Introduction

Line 72. Please, correct: “… are a major phenolic compound..”

Authors’ response: As observed by the reviewer, the phrase was corrected in the manuscript.

Since the work is a comparison among spent coffee grounds from different geographical origins, some data about the differences among the coffee from these regions should be presented. To evaluate the differences reported by the authors in this work, it is necessary to understand first if the species used are the same, as well as type (premium, traditional, etc). It is important to have these data in the introductory section

Authors’ response: As suggested by the reviewer, a section about the differences between Arabica coffees from the different origins has been added in the introduction, and some alterations were made in the Sample preparation section (3.2.) of Materials and Methods, regarding the reviewer’s comment.

Line 183. Since Figure 4 cannot fit a single page, please, bring part of this figure to page 5, to remove the empty space.

Authors’ response: Figure 4 was altered as suggested by the reviewer.

  1. Results and Discussion

Line 108. Please, add a reference and a brief explanation why µ - QuEChERS extraction procedure was chosen in this work.

Authors’ response: The reviewer’s suggestion was added to the manuscript.

The assays used (TPC, DPPH and ABTS) are very simple and, although highly used, they are not very specific and present several limitations, as the authors mention in the lines 133-134. To explain the choice of DPPH and ABTS assays, the authors say “...literature demonstrates that DPPH and ABTS are the most preferably em- 136 ployed synthetic radicals in antioxidant assays [28–30] …”. However, two of the references cited in this part (28 and 29) are from the group. In my opinion, these two references report the preference of the group by these assays, but do not represent the literature in general. Therefore, I recommend to replace these two references by references from authors which do not sign the current work.

Authors’ response: As mentioned by the reviewer, the references was replaced in the manuscript.

Lines 141-144: Here and in some other places, difference among the samples was detected, but the authors do not discuss the possible reasons for the difference. This is what I was talking about earlier; it should be highly desirable to add some data about the type of grounds used in the works, since there may be some characteristics that could explain the results. Even literature data could be used trying to explain or to hypothesize the reasons for the differences.

Lines 204-206: Please, add a reference to confirm that the method is acceptable.

Authors’ response: A reference was added to support the affirmation, as the reviewer’s suggestion.

Line 252. There is not Figure 20.

Authors’ response: The mistake pointed by the reviewer was corrected in the manuscript.

Line 265-268. Please, verify if the characteristics of the different grounds can explain the fact that caffeic acid and 1,5-CQA were found only in specific samples. In the discussion (lines 281-285 and then in the following lines), there is inference about the varieties (Arabica and Robusta), and this is an example of the type of discussion/comparisons that are missing in the manuscript (introduction and discussion).

Authors’ response: According to the reviewer’s suggestion, the information was added to the manuscript.

Lines 301 and followings. To have this section, it is necessary to give an ideia of the yield of the compounds in relation to commercial sources, because if the yield is very low in comparison with the current industrial sources, the use of spent coffee grounds is not competitive. I suggest to be clear if these residues are real potential sources of bioactive compounds; econnomically viable, I mean. In my opinion, it is not necessary to find high levels of caffeic acids and other compounds to have the work published, but I think that the results must be interpreted in order not to confuse the reader.

Authors’ response: According to the reviewers’ comment, the text was added in the manuscript.

Reviewer 2 Report

The author described and validated a new method for analysis of derivates of quinic acid derivates and hydroxycinnamic acids. The manuscript is well written and relevant literature is cited. Obtained results could be of great interest considering potential use of spent coffee grounds. 

I send you a list of my suggestions, comments and questions that could be considered for preparation of final form of your manuscript that would be acceptable for publication by my opinion.  

Material and Methods section

In methodology and methods, the description of sampling procedure should be improved. The authors do not provide any information about number of samples, whether composite samples were prepared for analyses, number of replicates for measurements. 

At the same time, the description of statistical analysis of data is not complete. Some relevant data are missing (statistical program that was used, how statistical significance of differences were tested and similar).

Results and discussion section

The results about polyphenols contents obtained with Folin-Ciocalteu procedure and UHPLC are not consistent. The authors reported almost 10 times higher values for quinic acid derivates determined by UHPLC then for total polyphenols. How the author explain obtained differences? Why they used two different methods for preparation of polyphenols extracts one for analysis of total polyphenols and antioxidant activities and the other for UHPLC? It seems that the content of polyphenols was not comparable in the extracts. Why did they analyzed water diluted extract after QuEChERS extraction? Do the authors except that the solubility of polyphenols left in spent coffee grounds in water is better than in methanol, ethanol or some other organic solvent? How the author explain no correlation between obtained total polyphenols and antioxidant activities? 

Presentation of results should be improved and mean value and standard deviation presented with the same number of digits (example 78.11+-7.333 replaced with 78.1+-7.3). Once again, the number of replicates of measurements reported.

General

I suggest to author that they add some pictures of quinic derivates structures and to add trivial names chlorogenic and neochlorogenic beside existing term that are used (example 3-caffeoylquinic acid, chlorogenic acid).

Best regards

Author Response

-The author described and validated a new method for analysis of derivates of quinic acid derivates and hydroxycinnamic acids. The manuscript is well written and relevant literature is cited. Obtained results could be of great interest considering potential use of spent coffee grounds.

I send you a list of my suggestions, comments and questions that could be considered for preparation of final form of your manuscript that would be acceptable for publication by my opinion.

Material and Methods section

In methodology and methods, the description of sampling procedure should be improved. The authors do not provide any information about number of samples, whether composite samples were prepared for analyses, number of replicates for measurements.

Authors’ response: As suggested by the reviewer, information about the samples and sampling procedure, as well as the number of replicates analysed, was added in the manuscript.

-At the same time, the description of statistical analysis of data is not complete. Some relevant data are missing (statistical program that was used, how statistical significance of differences were tested and similar).

Authors’ response: As mentioned by the reviewer, the information about the statistical analysis performed was added in the manuscript.

-Results and discussion section The results about polyphenols contents obtained with Folin-Ciocalteu procedure and UHPLC are not consistent. The authors reported almost 10 times higher values for quinic acid derivates determined by UHPLC then for total polyphenols. How the author explain obtained differences? Why they used two different methods for preparation of polyphenols extracts one for analysis of total polyphenols and antioxidant activities and the other for UHPLC? It seems that the content of polyphenols was not comparable in the extracts.

Authors’ response: We understand the reviewers comment regarding the use of two different methods for the Total Phenolic Content (TPC) and antioxidant activity and the UHPLC analysis and provide further explanation regarding this matter. µ-QuEChERS was chosen for the TPC and antioxidant activities because this methodology allows not only the extraction of the analytes of interest but also the removal of some interfering components, since it includes a clean-up step (in the present work the sorbents used in the clean-up were Primary secondary amine and magnesium sulfate, allowing the removal of sugars, fatty acids, organic acids, and anthocyanin pigments). The removal of these interferences minimizes the over-quantification when using spectrophotometric techniques. On the other hand, for the UHPLC analysis, a µSPEed methodology was employed because it permits a pre-concentration of the analytes of interest, when the extraction is performed on a matrix with low concentration of the analytes. Therefore, the TPC and UHPLC results are not comparable, since different extraction methods were applied.

-Why did they analyzed water diluted extract after QuEChERS extraction? Do the authors except that the solubility of polyphenols left in spent coffee grounds in water is better than in methanol, ethanol or some other organic solvent?

Authors’ response: The present aimed for the identification and quantification of bioactive compounds in the spent coffee grounds samples, and both extraction methods were performed as reported in literature with no optimization regarding extraction solvents being employed (the articles are referenced in the Materials and Methods section). How the author explain no correlation between obtained total polyphenols and antioxidant activities? Indeed, a correlation between the TPC and antioxidant activity should be expected, since phenolic compounds are potent antioxidants (samples with higher TPC value should present higher antioxidant activity). However, this was not observed in this work’s results. The authors believe this could be related to the presence of non-phenolic compounds in the extracts analyzed that possess antioxidant capacity (for instance, caffeine), and aid the reduction of the free radicals used. Since this is an important aspect that should be discussed, it was added in the manuscript in the Results and Discussion section.

-Presentation of results should be improved and mean value and standard deviation presented with the same number of digits (example 78.11+-7.333 replaced with 78.1+-7.3). Once again, the number of replicates of measurements reported.

Authors’ response: The mean values and standard deviation were corrected in the manuscript, as suggested by the reviewer.

-General I suggest to author that they add some pictures of quinic derivates structures and to add trivial names chlorogenic and neochlorogenic beside existing term that are used (example 3-caffeoylquinic acid, chlorogenic acid).

Authors’ response: The structures of the quinic acid derivates as well as some of their trivial names were added to the manuscript.

Round 2

Reviewer 1 Report

The manuscript was improved and, in my opinion, it can be accepted in the current form.

Reviewer 2 Report

The current form of manuscript is improved version due added changes in text. The authors replied to all questions, suggestions and comments that were send to them in review, The paper provides important scientific data about analysis of polyphenols present in coffee.

The manuscript can be accepted for publication in present form, by my opinion.